# Reduced efficacy of a Src kinase inhibitor in crowded protein solution

Kento Kasahara [1], Suyong Re [1], Grzegorz Nawrocki[2], Hiraku Oshima [1], Chiemi Mishima-Tsumagari[3], Yukako Miyata-Yabuki [3], Mutsuko Kukimoto-Niino [3], Isseki Yu[4], Mikako Shirouzu [3], Michael Feig [2✉] & Yuji Sugita [1,5,6✉]

The inside of a cell is highly crowded with proteins and other biomolecules. How proteins express their specific functions together with many off-target proteins in crowded cellular environments is largely unknown. Here, we investigate an inhibitor binding with c-Src kinase using atomistic molecular dynamics (MD) simulations in dilute as well as crowded protein solution. The populations of the inhibitor, 4-amino-5-(4-methylphenyl)−7-(t-butyl)pyrazolo [3,4-d]pyrimidine (PP1), in bulk solution and on the surface of c-Src kinase are reduced as the concentration of crowder bovine serum albumins (BSAs) increases. This observation is consistent with the reduced PP1 inhibitor efficacy in experimental c-Src kinase assays in addition with BSAs. The crowded environment changes the major binding pathway of PP1 toward c-Src kinase compared to that in dilute solution. This change is explained based on the population shift mechanism of local conformations near the inhibitor binding site in c-Src kinase.

[1] Laboratory for Biomolecular Function Simulation, RIKEN Center for Biosystems Dynamics Research, Kobe, Hyogo, Japan. [2] Department of Biochemistry and Molecular Biology, Michigan State University, East Lansing, MI, USA. [3] Laboratory for Protein Functional and Structural Biology, RIKEN Center for Biosystems Dynamics Research, Yokohama, Kanagawa, Japan. [4] Department of Life Science and Informatics, Maebashi Institute of Technology, Kamisadori-machi, Maebashi, Gunma, Japan. [5] Computational Biophysics Research Team, RIKEN Center for Computational Science, Kobe, Hyogo, Japan. [6] Theoretical Molecular Science Laboratory, RIKEN Cluster for Pioneering Research, Saitama, Japan. ✉email: mfeiglab@gmail.com; sugita@riken.jp

Understanding protein functions in a living cell is one of the essential issues in molecular biology and biochemistry. In vitro experiments on protein or enzyme functions often reveal different results from in vivo experiments. In a typical cytoplasm, 25–45% of the total volume is occupied by macromolecules including many proteins, nucleic acids, metabolites, osmolytes, and ions[1,2]. The crowded environment affects protein structure, stability, dynamics, and biological functions slightly or significantly[3,4]. Diffusion of a ligand toward a target protein is generally reduced upon crowding, whereas the transition-state stabilization and/or encounter complex formation are facilitated, as the protein concentration increases. These two effects make it difficult to understand the overall effect of crowding on enzymes[4]. Recent experimental[5–9] and theoretical studies[10–16] have shown that weak and non-specific molecular interactions are equally important with the excluded volume effect that has been emphasized in the traditional theory on macromolecular crowding[2,4]. For instance, in-cell NMR spectroscopy has shown reduced conformational stability of ubiquitin in Hela cells compared with those in dilute solution. The traditional theory, however, suggested that compact native structures of proteins are preferred in crowded cellular environments[7].

Substrate (or ligand) binding to a target protein is an essential process that controls a variety of biological functions including enzymatic reactions, substrate transports, signal transductions, and so on. The well-established concepts like "lock-and-key", "induced-fit", or "conformational selection" have been used to interpret the binding processes[17] as well as to design new drug candidates[18,19]. Protein conformational flexibility has been regarded as more important in substrate bindings so that atomistic molecular dynamics (MD) simulations are often conducted in in silico drug discovery[20]. In spite of these efforts, many designed drug candidates lack in vivo efficacy and cannot be applied to patients who are suffering from diseases. One of the missing factors in current drug discovery protocols might be the effect of cellular environments on protein–ligand bindings and functions. There is increasing experimental evidence that supports the importance of cellular environments for the thermodynamics and kinetics of protein–ligand binding[3,21]. For instance, the efficacy of inhibitors of threonine tyrosine kinases in cells is reduced significantly in many cases compared with that in dilute solution[22].

Recently, the accessible time scale of MD simulation has been greatly extended owing to the developments of MD-special-purpose supercomputers, Anton/Anton2[23,24], and enhanced conformational sampling algorithms including replica-exchange methods[25,26], metadynamics[27,28], Gaussian accelerated MD (GaMD)[29], and scaled MD[30,31]. Parallel MD simulations based on weighted-ensemble method[32–34], parallel cascade selection MD (PaCS-MD)[35], Markov state model (MSM)[36] are also useful to predict thermodynamic and kinetic properties of biomolecules. Using these advanced techniques, absolute binding free energies and/or kinetic parameters, namely, $k_{on}$ or $k_{off}$, in dilute solution have been successfully predicted. Atomistic MD simulations of multiple proteins in crowded environments revealed different aspects of proteins, such as translational or rotational diffusion[13,37], solvation structure and dynamics[12,14], and the conformational stability of proteins[10,12]. The importance of metabolites including adenosine triphosphate (ATP) on protein structures and interactions was suggested in large-scale MD simulations of the cytoplasm in *Mycoplasma genitalium*[15]. Independent experiments also suggested that ATP works as a hydrotrope in cells[38].

The c-Src kinase (Supplementary Fig. 1a) regulates various signal transduction pathways by catalyzing phosphate transfer from ATP to a target substrate[39]. Dysregulation of this kinase function is associated with many diseases like cancer, making it an important therapeutic target[40,41]. To design drug candidates, both the active and inactive conformations of c-Src kinase[42–47] as well as the ligand-binding process[45,48–50] have been extensively investigated in dilute solution. PP1 was designed to inhibit the kinase function by binding to the ATP-binding site (the canonical binding site) of c-Src kinase (Supplementary Fig. 1b)[51]. A recent computational study using the two-dimensional replica-exchange method characterizes the multiple binding poses and binding pathways of PP1 to c-Src kinase from the encounter state[50,52].

In this work, we study protein functions in crowded protein solutions by performing multiple μsec MD simulations using Anton2 and other supercomputers (~100 μsec, in total). We focus on a small inhibitor (4-amino-5-(4-methylphenyl)−7-(t-butyl) pyrazolo[3,4-d]pyrimidine, PP1) binding to c-Src kinase in dilute solution and crowded environments with bovine serum albumins (BSAs) as protein crowders (BSA has been shown not to change the conformational stability of the other protein in crowded solution[6]). We perform atomistic MD simulations of four c-Src kinase/PP1/BSA systems, each of which contains 0 (dilute solution), 2, 4, and 8 BSAs in a simulation box with the same size (referred to as Src2BSA, Src4BSA, and Src8BSA, respectively). The observations in the simulations provide a microscopic picture of protein-inhibitor binding in crowded solutions, deepening our understanding of protein functions in a living cell and proposing a new strategy for in silico drug discovery.

## Results

**Structural stability and flexibility of c-Src kinase**. We first examine the effect of crowded environments on the structural stability and flexibility of c-Src kinase. The c-Src Cα root mean square deviations (RMSDs) with respect to the X-ray structure are calculated to be c.a. 2–3 Å in all the simulations (Supplementary Fig. 2). The time averages of the RMSD values in the presence of BSA (~2.4 Å) are slightly smaller than that in the absence of BSA (~2.6 Å) (Supplementary Fig. 3a). The Cα root mean square fluctuations (RMSFs) of the kinase in the inhibitor-unbound state are also not affected by the presence of the BSAs (Supplementary Fig. 3b), where the unbound state is defined when the protein-PP1 distance, $\xi$, is longer than 15 Å (Supplementary Fig. 1d). Two pseudo-torsion angles of Asp-Phe-Gly (DFG) motif (Supplementary Fig. 4), which represent active/inactive states of the kinase[53], show that the kinase remains in the active-like initial conformation (DFG-in and αC-out state) during the simulations.

**PP1 distributions in water and on the protein surfaces**. The interaction between c-Src kinase and PP1 was examined based on the spatial distribution function (SDF) of PP1 around the kinase. The SDF in the absence of BSA illustrates the existence of many non-canonical binding sites beside the canonical ATP-binding one, which is consistent with previous findings in long MD simulations using Anton[54] (Fig. 1a and Supplementary Fig. 5). The PP1 non-canonical binding sites eventually disappear as the BSA concentrations increase. The surface region of c-Src kinase in the crowded solutions is largely occupied by BSAs (Fig. 1b and Supplementary Fig. 6). Since the canonical binding site exists at the cleft formed between the N- and C-lobes, BSA cannot cover it. Many PP1 stay on the surface of BSA in the crowded solutions, owing to weak and non-specific interactions between PP1 and BSAs (Supplementary Fig. 7a). The binding sites of PP1 on BSA resemble those of other compounds found in the crystal structure (PDB ID: 4JK4), reflecting the general feature of BSA as a vehicle of small molecules in the circulatory system (Supplementary Fig. 7b)[55]. The reduced PP1 distribution around c-Src kinase in the crowded solutions is, thus, explained via both the steric

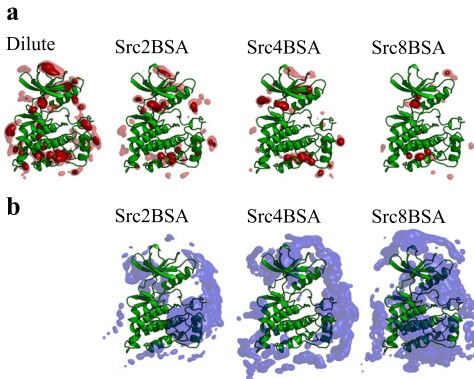

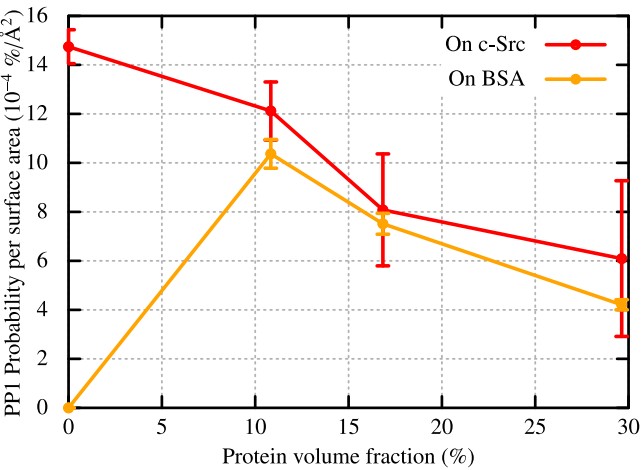

**Fig. 1 Decrease of non-canonical PP1-binding sites upon crowding.** Spatial distribution functions (SDF) of **a** PP1 and **b** BSA around c-Src kinase. In the case of BSA-SDF, Cα atoms are used for analysis. The PP1-SDF is shown as isosurface at 0.5% (transparent) and 1.5% occupancies (solid). For BSA-SDF, the isosurface at 1.0% occupancy is shown.

**Fig. 2 The probability of finding PP1 per unit protein surface area (Å²) on the surface regions of c-Src kinase and BSA.** The probability is estimated from the solvent-accessible surface area (SASA) of the crystal structures of c-Src ($S_{Src}$= 13630.75 Å²) and BSA ($S_{BSA}$= 26680.10 Å²). The SASAs are calculated with the Spatial decomposition analysis (SPANA) tool[78]. Error bars indicate the standard error of the mean from seven trajectories for each system.

hinderance of BSAs (Supplementary Fig. 6) and the weak and non-specific interactions between PP1 and BSAs (Supplementary Fig. 7a).

We also investigated the population of PP1 in the bulk water region, which is at least 5 Å away from any heavy atom of c-Src kinase or BSAs (Supplementary Fig. 8). The space within 5 Å of any heavy atom of c-Src kinase (or BSA) is defined as the surface region of c-Src kinase (or BSA). The probability of finding PP1 on either c-Src kinase or BSA surface (per unit surface area) is shown in Fig. 2. The probability of c-Src kinase decreases from $14.7 \times 10^{-4}\%/\text{Å}^2$ (dilute) to $6.1 \times 10^{-4}\%/\text{Å}^2$ (Src8BSA). A similar decrease in the probability for PP1 on BSA surface with increasing the crowder concentration (from $10.4 \times 10^{-4}\%/\text{Å}^2$ in Src2BSA to $4.2 \times 10^{-4}\%$ in Src8BSA) was also observed, indicating that both the kinase and BSAs competitively capture PP1s in the crowded condition. This observation is consistent with the PP1-binding free energies at different binding sites on c-Src kinase and BSA estimated through molecular mechanics-generalized Born surface area (MM-GBSA) analysis (Supplementary Fig. 9). The range of binding free energies at different c-Src kinase-PP1-binding sites are comparable to those

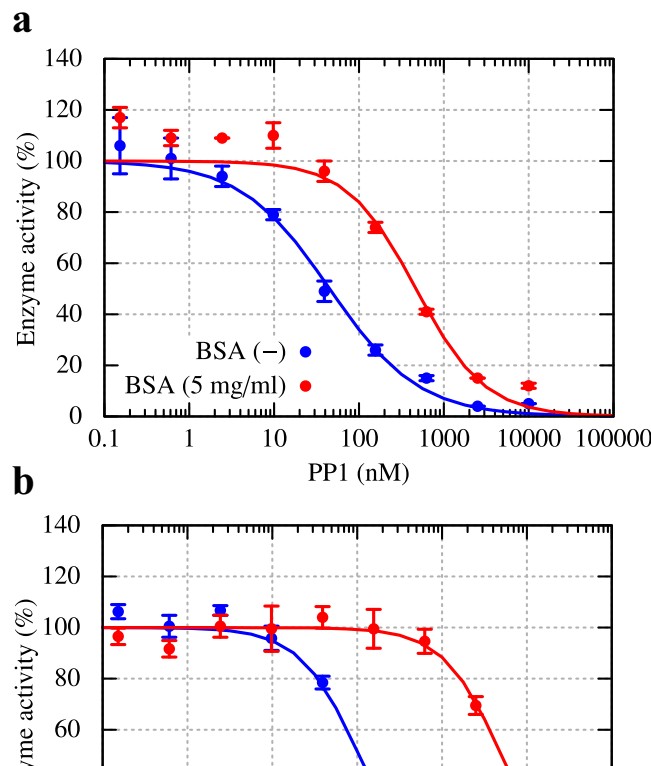

**Fig. 3 Reduced efficacy of PP1 inhibitor in the presence of the crowder (BSA) proteins. a** Enzymatic activity of c-Src kinase in the dilute solution ([c-Src]=65.7 nM) and in the crowded solution with BSA (5 mg/ml). **b** The same experiment except for the concentration of BSA (100 mg/ml) in the crowded solution and those of DMSO (0.2% and 1.0% in the first and second experiments, respectively). In both crowded solutions, the concentration of c-Src kinase is [c-Src]=65.7 nM. Error bars represent the standard deviation of the mean from three independent assays, respectively.

at BSA-PP1-binding sites. Owing to the crowding, the probability of finding PP1 in the water phase is significantly reduced (Supplementary Fig. 10, Supplementary Table 4). In the dilute solution, 79.9% of PP1 exists in the water phase, while the probability drops to 1.8% in Src8BSA.

**In vitro inhibitory assays in the presence or absence of BSA.** In vitro, PP1 inhibition assays with c-Src kinase, Src substrate, and ATP, were performed in the presence or absence of BSA to test this hypothesis. To avoid aggregation in the protein solution, the protein concentration in the experiments was much lower than in the simulations. The c-Src kinase concentration was set to 65.7 nM in the absence of BSA, whereas the BSA concentration was much larger (12.7 μM = 5 mg/ml) than kinase (65.7 nM) to realize a crowded protein solution. The 50% inhibition concentration (IC$_{50}$) of the PP1 considerably increased from 45.1 nM to 470 nM in the presence of BSA (Fig. 3). We also performed the same inhibition assays at higher concentrations of BSA (100 mg/ml). In this condition, the enzyme activity was reduced to ~30% at [PP1] = 10 μM and higher concentrations of PP1 were difficult to

achieve because of the limited solubility of PP1. The resulting $IC_{50}$ of PP1 further increased to 5.03 μM, indicating even weaker inhibition by PP1 in the presence of higher concentrations of BSA. The concentrations of DMSO in the first (Fig. 3a) and the second (Fig. 3b) experiments are slightly different (0.2% vs 1.0%, respectively). However, they did not significantly affect the efficacy of PP1 in dilute solution compared with the crowding effects. This validates our hypothesis of a reduced inhibitor efficacy in crowded protein solutions.

**Slowdown of PP1 diffusion in crowded solution.** The mean square displacements of PP1 were analyzed in the bulk water region, on the surfaces of c-Src kinase or BSA. PP1 diffusion in the bulk water region slowed down as the protein volume fraction increased (Supplementary Fig. 11a). PP1 diffusion on the protein surfaces is nearly identical to the diffusion of c-Src kinase or BSA in Src4BSA and Src8BSA (Supplementary Fig. 11b, c). In the crowded solutions, the probability of finding PP1 on the protein surfaces is much higher than that in the bulk water region (Supplementary Fig. 10). PP1 diffusion is thus significantly affected by proteins in the crowded solutions as PP1 mostly diffuses along with the proteins while bound to it. This is further manifested by long residence times on the 1–100 ns time scale for PP1 association with the c-Src kinase and BSA as shown in Supplementary Table 6.

**Ligand-binding pathways in crowded environments.** In the present simulations, we observed four and two PP1-binding events to the canonical binding site in the dilute solution and Src8BSA, respectively (Supplementary Figs. 12 and 13). In dilute solution, PP1 approaches the canonical binding site from the glycine-rich loop (G-loop) in c-Src kinase. This binding pathway is different from that observed in Src8BSA: PP1 first makes a contact with either the N-terminal side of G-loop or hinge region (Supplementary Fig. 13) and then intrudes into the canonical binding sites. These binding trajectories are projected onto the two free energy landscapes (FELs) that were obtained in our previous study on the same system in dilute solution using the gREST/REUS simulations[50] (Fig. 4 and Supplementary Figs. 14–16). The two-dimensional FELs illustrate the changes in free energy along with the c-Src kinase-PP1 distance ($\xi$) and one of the coordinates (an azimuth angle $\theta$) representing the PP1 position with respect to c-Src kinase (Fig. 4a). Two FELs were computed in two different conditions: in one simulation, c-Src kinase fluctuated freely (referred to as "free") (Fig. 4b and Supplementary Fig. 14a), whereas strong positional restraints to the kinase prohibited the kinase flexibilities in the other simulation (referred to as "restraint") (Fig. 4b and Supplementary Fig. 14b). On the major binding pathway in the "free" FEL, PP1 approaches c-Src kinase from the small $\theta \sim 60°$ (Supplementary Fig. 14a), while the binding pathway of the "restraint" FEL starts from the large $\theta \sim 120°$ (Supplementary Fig. 14b).

The binding trajectories in dilute solution fit to the major pathway of "free" FEL (Fig. 4c) and those in Src8BSA fit the major pathway of "restraint" FEL (Fig. 4d). The major difference in two pathways is the interaction at an initial encounter state: PP1 interacts with the G-loop in dilute solution (**E**) (Supplementary Fig. 12), whereas it interacts with the N-terminal side of the G-loop or hinge region in Src8BSA (**E**$_r$) (Supplementary Fig. 13). To examine if the difference happened to be observed, we performed additional 30 simulations (each for 20 ns) in Src8BSA starting from **E** and **E**$_r$ states. Although the simulations from **E**$_r$ state lead to three binding events, no binding was observed in the simulation from **E** state. Similar to the binding trajectories shown in Figure 4d and Supplementary Fig. 16, the

binding trajectories from **E**$_r$ state fit to "restraint" FEL (Supplementary Fig. 17). This suggests that the binding pathways might be altered in the presence of crowders.

**Conformational shifts of a Tyr sidechain upon crowding.** To examine the molecular mechanisms for why the binding pathways differ between dilute and crowded solutions, we investigated local conformational changes in c-Src kinase as protein volume fraction increased. We found two characteristic sidechain conformations of Tyr82 at the hinge region, which we refer to as TYR-in and TYR-out. In TYR-in, two hydrophobic residues Leu15 and Gly86 sandwich Tyr82, thereby blocking the pathway from the hinge region (Fig. 5a). PP1 thus interacts with G-loop to form the encounter complex **E**. In TYR-out, on the other hand, G-loop shifts downward (~1 Å, Supplementary Fig. 18). Consequently, Tyr82 is positioned away from Leu15 and Gly86 (Fig. 5b), allowing PP1 access to the hinge region at the encounter complex **E**$_r$. The distribution of Tyr82-Gly86 distance, $g(d)$, is centered around $d = 5$ Å in TYR-in and $d = 8.5$ Å in TYR-out (Fig. 5c). TYR-in is dominant in the dilute condition, while we observe TYR-out as a minor population. As the protein concentration increases (in the order of Src2BSA, Src4BSA, and Src8BSA), the ratio between TYR-in and TYR-out is gradually changed. Finally, TYR-out becomes the major population in Src8BSA. The additional 1 μs-simulations using a different force field, the CHARMM36m force field[56], support that the observed population shift between TYR-in and TYR-out is universal (Supplementary Fig. 19).

In a crystal structure of c-Src kinase in complex with an inhibitor (PDB ID: 1Y57), the Tyr82-Gly86 distance is close to, but slightly different from that in TYR-out (Fig. 5c). Interestingly, a water molecule exists between Tyr82 and Gly86 in the crystal structure of c-Src kinase in the apo form (PDB ID: 1YOJ) (Supplementary Fig. 20). In the same position, we observe water densities in the MD snapshots where the Tyr82 sidechain assumes the TYR-out structure.

## Discussion

The crowding effects on c-Src kinase/PP1 binding can be categorized into two different types: (1) The slowdown of PP1 diffusion, the decrease of non-canonical PP1-binding sites in the kinase, and the reduced efficacy of the inhibitor are due to generic crowding effects. We expect similar effects with other proteins under crowded conditions. (2) The different PP1-binding pathways in crowded vs. solution are specific to c-Src kinase. Whether such an effect is present for other ligands and other enzymes is highly dependent on the molecular details of a given system. Hereafter, we reconsider these two crowding effects based on the traditionally excluded volume effects as well as weak and non-specific molecular interactions in crowded and cellular environments.

The slowdown of PP1 diffusion in crowded solutions is consistent with the traditional understanding of crowding, which predicts that the kinetics of protein–ligand binding becomes slower under crowded conditions due to increased solution viscosity[3]. Indeed, we found that the translational diffusion of PP1 and water even in the bulk water region of the crowded solutions become slower (Supplementary Fig. 21). On the protein surfaces, ligand diffusion was much more retarded due to non-specific interactions with proteins. In our previous MD simulations of the *MG* cytoplasm, charged and/or large metabolites were also found to stay on the protein surfaces for a long time, whereas hydrophobic metabolites like Val were found to remain solvated in larger fractions[15]. The importance of non-specific interactions between macromolecular crowders and substrates was also seen

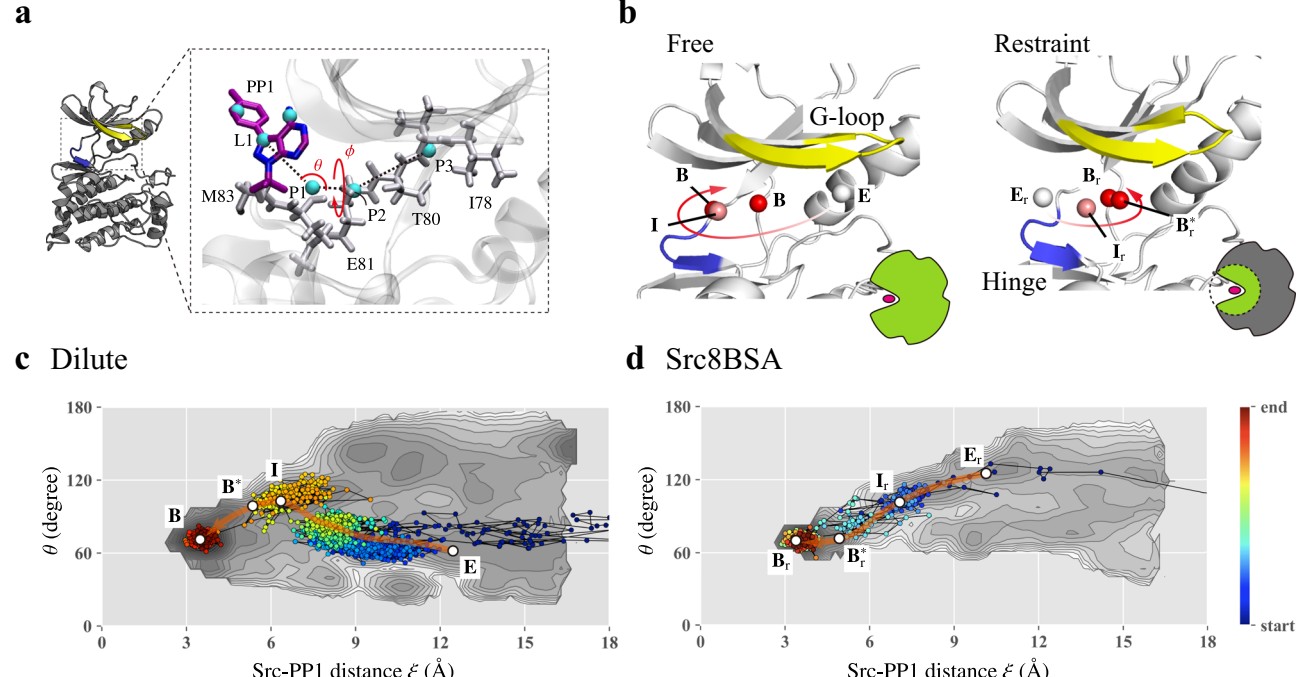

**Fig. 4 Binding trajectories projected onto the two-dimensional free energy landscapes (FELs) obtained from the gREST/REUS simulations in the dilute solution. a** Definition of azimuth angles ($\theta$, $\varphi$) for describing the relative position (See Supporting information for the detail). **b** The major binding pathway in the dilute solution obtained from the gREST/REUS simulation in different conditions. Left panel: simulation without any restraints (free), Right panel: simulation in which the residues outside the binding pocket are kept close to the crystal structure. **E**, **I**, **B***, and **B**, respectively, indicate encounter complex, intermediate, pre-bound, and bound states. The subscript r denotes the presence of the restraints. **c**, **d** The binding trajectories in the dilute solution and in the crowded solution (Src8BSA) projected onto the 2D-FEL from "free" simulation and that from "restraint" simulation, respectively.

in a recent enzyme assay[57]. Therefore, it is becoming evident that the heterogeneous environments for proteins and ligands in crowded cellular conditions must be taken into account to correctly estimate protein–ligand-binding kinetics in a living cell. On the computational side, this insight could be incorporated into Brownian dynamics-based approaches[58–61], which are commonly used to predict protein–ligand-binding kinetics for dilute conditions. The traditional theory of macromolecular crowding predicts that protein–ligand and protein–protein interactions in crowded environments would be stronger than those in dilute solutions. The volume exclusion effect has been considered as a main driving force[21,62]. This effect suggests an increase in the effective concentration of the substrates around the target macromolecule. For example, the activity of a DNA ligase correlates with an increase in the effective concentration of the substrate due to crowding[63]. Contrary, the present study shows that the effective ligand concentration is reduced as a function of protein volume fraction because target protein-crowder interactions sterically block non-canonical binding sites. Ligands might be trapped on the surfaces of crowder proteins. This is again consistent with the observations in our previous simulations of the *MG* cytoplasm.[15] Based on these common observations, we hypothesize that an effectively reduced ligand concentration in soluble regions is common in crowded solutions as well as cellular environments like the cytoplasm. A decrease in the ligand concentration is expected to reduce the inhibitor efficacy in these environments, as our kinase assay showed. It should be also mentioned that increasing concentrations of crowders, as well as inhibitors, can cause artifacts, the precipitation of inhibitors for instance, undesired for improved drug efficacy. In future work, binding free energy calculations under crowded conditions could quantitatively address the relation between the effective inhibitor concentration and its efficacy.

In Src8BSA simulations, the PP1-binding pathway is entirely different from that in dilute simulations. This suggests that crowding effects could change protein–ligand-binding mechanisms. Since we only perform conventional MD simulations, the sampling of binding events is still low. Further investigation with enhanced sampling techniques in the presence of the crowders could further clarify the change of the pathway due to the crowding. The crystal structure of c-Src kinase has a local structure similar to what is dominant in Src8BSA. However, the crystal environments and crowded solutions are different conditions in terms of their flexibilities and molecular interactions with nearby molecules. We observed an increasing shift in the population of the local conformation of c-Src kinase as the BSA concentration increases. As shown in Supplementary Fig. 22, the transitions between TYR-in and TYR-out forms occur dynamically. Even in the dilute solution, TYR-out exists as a minor conformation. Owing to non-specific molecular interactions between crowder BSAs and the G-loop of c-Src kinase, TYR-out conformations are preferred in the crowded environment (see Supplementary Fig. 18).

In the study, we observed binding events only in the dilute and Src8BSA systems, which are considered as two extreme conditions. These two conditions may increase the possibility of binding processes on different pathways. Since the conformational changes between TYR-in and TYR-out happen as population shifts (Fig. 5), longer MD simulations in Src2BSA and Src4BSA could increase the possibilities of ligand-binding processes through one of the two pathways. Therefore, in addition to the macromolecular crowding effects on open-to-close conformations or folding/unfolding equilibrium[64,65], small and local conformational shifts near the active site of proteins due to crowding may change molecular mechanisms underlying their biological functions as we describe in this study. To examine the

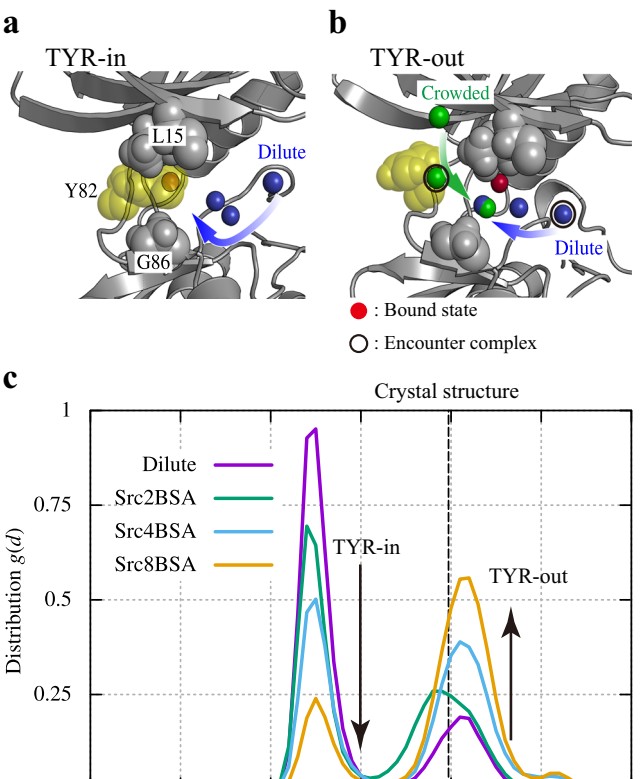

**Fig. 5 Distribution of the TYR-in and TYR-out states of c-Src kinase.**
**a** TYR-in state structure together with a binding trajectory in the dilute solution. **b** TYR-out state structure with binding trajectory in the dilute and crowded solution (Src8BSA). Red, blue, and green spheres represent the PP1 bound position in the crystal structure, representative binding pathways in dilute and crowded solutions, respectively. Encounter states in the pathways are circled. **c** Distribution for Tyr82-Gly86 distance, $g(d)$. The distance between Tyr82 and Gly86 in a crystal structure of c-Src kinase in a complex with an inhibitor (ID: 1Y57) is shown with dashed line for comparison[68].

difference of c-Src kinase-BSA interactions between TYR-in and TYR-out conformations, we calculated the average number of the Cα atom contacts between BSA and c-Src kinase for the two conformations individually (Supplementary Fig. 23). However, no significant differences between the two conformations are observed, suggesting that the crowder effects on the local protein conformation could be more complicated. The effects of different crowder types, for instance, the size of crowders[6] on protein structures and stability were investigated experimentally. The intracellular environments indeed consist of heterogeneous proteins and other biomolecules, as we simulated in the previous simulation study[15]. A further investigation using heterogenous crowders along the present study would lead to a deeper understanding of how the local conformational changes in more realistic cellular environments alter the binding mechanism.

We showed that the sidechain position of Tyr82 driven by the downward shift of the G-loop is the key determinant for changing the PP1-binding pathway. It has been suggested previously that the downward shift of the G-loop could impact drug efficacy[22]. Most of the apo form crystal structures of kinases, including c-Src kinase (PDB ID: 1YOJ)[66], have TYR-out conformation, although there are crystal structures having TYR-in conformation (such as protein kinase B/Akt[67]). The sequence alignment data set of 490 kinases[53] (Supplementary Fig. 24) shows that ~44% of kinases

share the tyrosine near the hinge region. In addition, ~16% of the kinases have phenylalanine instead of tyrosine at the same position. Therefore, the suggested role of Tyr82 as a key amino acid in the binding of ligands and inhibitors—and the possibility to be sensitive to crowding effects suggested here—may be shared by a broad range of kinases.

## Methods

**MD simulations.** The initial structure of c-Src kinase was prepared from an X-ray structure of unphosphorylated active-like c-Src kinase (PDB: 1Y57)[68]. Similar to the previous conventional MD[54] and the gREST/REUS simulations[50], only the kinase domain (residues 82-258) was used in this study. A c-Src kinase and 24 mM PP1 inhibitors were placed and then solvated by 150 mM NaCl aqueous solution (dilute). To represent crowded protein solutions, BSA proteins (PDB: 4F5S)[69] are also included as crowders. In Src2BSA, Src4BSA, and Src8BSA, two, four, and eight BSAs were simulated with c-Src kinase and PP1. We used the AMBER ff99SB-ILDN[70,71] and TIP3P[72] parameters for the proteins and water molecules, respectively. The PP1 parameters were prepared with GAFF with AM1-BCC[73]. For each system, a set of different initial configurations was prepared. The simulations were performed with the GENESIS software[74,75] and with Anton2[24] at the Pittsburgh Supercomputing Center. Further detail is available as a part of Supplementary Information.

**Protein expressions and purification.** The gene encoding human Src kinase domain (residues 260–533) was cloned into the pCR2.1 vector (Invitrogen) and expressed as a fusion with N-terminal histidine and GST tags using *Escherichia coli* cell-free reaction supplemented with chaperones and YopH[76,77]. The protein was purified by affinity chromatography using a HisTrap column (GE Healthcare) and further by ion exchange on a HiTrap Q column (GE Healthcare) and used for in vitro kinase assay.

**In vitro inhibition assay.** In vitro inhibition assay was performed using the ADP-Glo Kinase Assay (Promega) under the conditions based on the manufacturer's protocol. In brief, purified Src protein (20 ng) was incubated with the indicated concentrations of PP1 (Cayman Chemical), 100 μg/ml Src substrate KVEKIGEGTYGVVYK-amide (SignalChem Pharmaceuticals), and 10 μM ATP for 60 min at room temperature in a reaction solution containing 0.2% DMSO, 40 mM Tris-HCl (pH7.5), 20 mM MgCl$_2$, 2 mM MnCl$_2$, 50 μM DTT with and without 5.0 mg/ml BSA. The reaction was terminated by adding the ADP-Glo Reagent, and then ADP generation was detected using the luciferase reaction by monitoring the luminescence on an ARVO X3 microscale luminometer (PerkinElmer). The enzyme activity in the absence of PP1 was set to 100% in each condition. IC$_{50}$ was determined by curve fitting the data using the GraphPad Prism8 program (GraphPad software). The fitting function is given by

$$\text{(Enzyme activity)} = \frac{100}{1 + \left(\text{IC}_{50}/[\text{PP1}]\right)^\alpha}, \quad (1)$$

where $\alpha$ is Hill coefficient. IC$_{50}$ and $\alpha$ are used as the fitting parameters, and the constraint was imposed on the fitting so as to satisfy the enzyme activity of 100% in the absence of PP1. The second in vitro inhibition assay was performed in the same conditions except for the concentrations of BSA (100 mg/ml) and DMSO (1.0%).

## Data availability
The data that support the findings of this study are available from the corresponding author upon reasonable request. Source data are provided with this paper.

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

## Acknowledgements

Grant or contribution numbers may be acknowledged. We acknowledge the computing time granted by RIKEN Advanced Center for Computing and Communication (HOKUSAI GreatWave and BigWaterfall), RIKEN Center for Computational Science (K computer), and HPCI system (Project ID: hp170254, hp180201, hp180274, hp190097, hp190181, hp200129, hp200135). This research was supported by MEXT as "Priority Issue on Post-K computer" (Building Innovative Drug Discovery Infrastructure Through Functional Control of Biomolecular Systems) (to Y.S.), "Program for Promoting Researches on the Supercomputer Fugaku" (Biomolecular dynamics in a living cell/MD-driven Precision Medicine) (to Y.S.), MEXT Grant-in-Aid for KIBAN (S) (19H05645) (to Y.S.), RIKEN pioneering projects "Dynamic Structural Biology", "Glycolipidologue Initiative" (to Y.S.) and "Biology of Intracellular Environments" (to Y.S. and M.S.). Funding was provided by the National Science Foundation grant MCB 1817307 (to M.F. and G.N.), by the National Institutes of Health grant R35 GM126948 (to M.F.). Computer time was used on the Anton2 special-purpose supercomputer at the Pittsburgh Supercomputing Center (PSCA18053P). We thank Mr. Kazuharu Hanada and Dr. Kazushige Katsura for their technical assistance on the experiments.

## Author contributions

K.K., G.N., and H.O. performed simulations. K.K. analyzed data. K.K., S.R., and I.Y. modeled simulation systems. C.M.T., Y.Y., M.K.N., and M.S. carried out experiments. K.K., S.R., M.F., and Y.S. wrote the manuscript. M.F. and Y.S. designed and initiated the research.

## Competing interests

The authors declare no competing interests.
