## [Peer Review File · Nature Communications]

REVIEWER COMMENTS

Reviewer #1 (Remarks to the Author):

The manuscript is an important contribution and an exciting piece of science. Since the pioneering computational efforts carried out by the authors teams (M. Feig's and Y. Sugita's), this new work is a beautiful example showing that the modelling of the macromolecular crowding is an essential aspect to consider in order to understand *in vivo* biological processes.

The manuscript is clear, well written and presented. The technical aspects are neat. The *in-silico* results, obtained with great statistics considering the systems, are complemented by *ad hoc* experiments that try to reinforce the main conclusion of the investigation. I consider the manuscript publishable in *Nature Communication* but several aspects need to be addressed prior to publication. The points are listed below.

My first aspect concerns the "sequestration" thermodynamics of the inhibitor that is at the origin of the trapping mechanism of the molecule on the surface of the crowders. This interaction limits the independent diffusion of the molecule in the solvent 'phase', producing a strong slowdown that in turn quenches the access to the target protein. It will be important to report/show/stress that within the used force field the solvation free energy of the ligand and the interaction free energy with representative surface patches of the crowder BSAs are not unbalanced artificially. The authors can easily produce these data by independent calculations.

Strictly related to the point mentioned above, the authors should better discuss and account for the discrepancy between the simulated and the experimental conditions. In the experiments, they only add 5 g/L of BSA (which corresponds to a very small volume fraction of ~ 0.004 , right?). So, to observe any strong difference in the bulk concentration of the inhibitor, which they suggest as a main rationale of the decreased inhibitor efficiency, the inhibitor should bind very strongly to BSA. However, in the simulations, they only observe a drop in the bulk concentration of the inhibitor from 80 % in dilute conditions to 30 % at a volume fraction of 0.11. This says that for the much smaller experimental volume fraction, the bulk concentration should be very similar to the dilute case. Therefore, the sequestration would not be sufficient for explaining the experimentally detected order-of-magnitude shift in inhibitor concentration.

On the basis of the sequestration on crowders surfaces and the associated slowdown dynamics of the inhibitor, the authors open a discussion about drug design strategy in order to account for the realistic environmental conditions of the cell interior. Here I have a question. Looking at the experiments reported in Fig. 2 bottom panel it seems that in order to recover the inhibition efficiency observed in dilute condition it suffices to increase the inhibitor concentration (a factor of 10). In drug design and subsequent essays, the concentration is always a free parameter that needs to be tuned to grant efficiency and avoid toxicity. If the author enters in this arena, they should for instance discuss what concentration limits should be avoided. Is there an example that can be reported? Given the great experience cumulated with past simulations (see the authors past works), is there a rule of thumbs that relates slowdown motion & local sequestration and crowder concentration so to estimate the possible shift in drugs concentration needed to recover dilute solution efficiency?

Finally, a personal view. Considering that great effort in drug design is spent to refine the chemistry of the active molecule to grant high affinity with target binding sites, and that basically the slowdown/sequestration caused by the environment seems not highly specific, I think very difficult to incorporate these two requirements (highly affinity for a target, poor interactions with general protein surfaces) at once. Have the authors some ideas? A finest analysis of the inhibitor with the crowders, that is not present in the manuscript and should, could provide some suggestions. I think also the authors should add in the introduction a note on the effect of diffusion in molecular recognition under crowding. The competitive effect of local high concentration and slowdown motion reported in the "Discussion" section, can be already anticipated in the introduction.

One extra remark concerns the discussion of the conformational changes induced by the BSAs crowders and the altered distribution of the Y82-G86 distance, a proxy for the conformational state of Tyr82. To me, the text seems light on this. The authors observe a change but I do not find a connection with the presence of the crowders. I think this must be better described/analyzed and a structural effect indicated, or at least questioned.

One delicate aspect the manuscript does not address is the homogenous vs heterogeneous crowding effect. Since we know that big vs small crowders can have different effects on local packing, stability, and conformational fluctuations, and since the change in probability of Tyr82 is visible even in presence of two BSAs, the authors could, with a small effort, produce a simulation with smaller crowders, or replacing 1 BSA with a different crowder. This would reinforce the manuscript and the results.

Final curiosity. It is intriguing that binding events are visible only in the simulations of the dilute system and at the highest crowding concentration, and not at intermediate values of crowding. See Table 1 in SI. Why? This must be discussed!

The paragraph reporting on binding pathway is present twice in the manuscript (pg 9 lines 176-190, and pg 10 191-205). A cut and paste error, I guess.

Fig. 2. The volume fraction associated to the experimental condition, BSA 5 mg/ml, should be indicated so to have a link between the top and the bottom panels.

Fig. 3 can be improved by labeling explicitly panels c and d, with the "dilute" and "Src8BSA" labels so to ease a reader to spot the differences immediately.

Reviewer #2 (Remarks to the Author):

The authors investigated the diffusion of the low molecular weight inhibitor PP1 and its binding to Src kinase in the absence and presence of crowding. Crowding is mimicked by the use of different concentrations of the crowder bovine serum albumin. The authors suggest, based on experiments, that PP1 is less efficient as an inhibitor in the presence of crowders. They also observe that PP1 is less available in solution with increasing crowder concentration, and that crowders may change the pathway for ligand binding due to conformational changes in Tyr82 of Src kinase.

The manuscript is generally clearly presented and mostly well written. The manuscript addresses a significant problem for which molecular simulations can give important insights. The results are original and interesting. However, I have some major concerns as detailed below, some details are missing in the methods, and the conclusions are not fully supported by the results.

Major concerns

1- Low sampling of binding pathways

Page 14, line 269: 'In Src8BSA simulations, the PP1 binding pathway is entirely different from that in dilute simulations.'

The authors obtained very few binding events in the simulations, especially in crowded conditions (4 and 2 events in the absence and presence of crowding, respectively). Therefore, they would need to sample more binding events to be able to claim that binding pathways are indeed different in the absence and presence of crowding.

2- Experimental results

The plot in Fig 2a shows computed values that do not directly relate to the main heading of the figure as a function of crowder protein volume fraction. The corresponding experimental value for the protein volume fraction would be at less than 1% and at a different ratio of kinase to crowder. Thus, this plot has little relation to the experimental data in Fig 2b and I think it would be better to put these two plots into different figures.

Page 8, lines 151-153: 'The 50% inhibition concentration (IC₅₀) of the PP1 considerably increased from 33.80 nM to 290.4 nM in the presence of BSA (Fig. 2(b)). This validates our hypothesis of a reduced inhibitor efficacy in crowded protein solutions.'

Here, the authors need to point out that the enzyme activity is higher in the presence of BSA and give what the reference was for defining 100% enzyme activity. If the reduced inhibitor efficacy is due to differences in inhibitor-protein binding, one wonders why the same effect is not observed for substrate – indeed, the data suggest the opposite. However, at no point is it stated what the Src substrate measured actually is (I309, p15). This information needs to be added, along with data on the dependence of Src substrate binding on BSA crowding, e.g. from analogous simulations to those for the inhibitor.

3- Incomplete description of computational methods

The setup and parameters for simulations are well described, but there is no information (in the manuscript or in the supporting information) about how the analysis of these simulations was performed (calculation of spatial distribution functions, free energy landscapes, and mean square displacements; definition of the bound state). This should be added.

4- Page 2, lines 44-45: 'Protein functions in a living cell could be examined using atomistic MD simulations with realistic cellular environments.'

This is an overstatement, compared to what is presented in the paper and must be removed or edited. The interaction with one artificial crowder, BSA, is studied.

Minor concerns

5- Abstract: Give the full name of PP1. It only becomes clear what it is in Supplementary Figure 2 and this figure requires a pdb identifier and a citation for the crystal structure. "imposed between the highlighted carbon atoms in PP1s.". Only 1 carbon is highlighted by a red circle. Rephrase this sentence to avoid confusion.

6- Page 4, line 77: 'Atomic MD simulations' – change to atomistic or atomic detail

7- Page 4, lines 89-90: 'Dysregulation of this kinase function is associated with many diseases like cancer, making it an important therapeutic target.'

A reference for this statement is missing.

8- Page 4, lines 95-96: 'We performed atomistic MD simulations of four c-Src kinase/PP1/BSA systems, each of which contains 0 (dilute solution), 2, 4, and 8 BSAs in a simulation box'.

What would be the corresponding concentration of BSA in the simulations, in g/L? This should be added to Supplementary Figure 3.

9- Page 5, line 105: 'The Ca root mean square deviations (RMSDs) from the X-ray structure are observed around 2-3 Å in all the simulations, indicating that the kinase remains in the active conformation'.

A small RMSD does not indicate whether the kinase is in the active conformation or not. The authors should examine the residues in the DFG motif to make this statement.

10- Page 5, lines 106-108: 'The time averages of the RMSD values in the presence of BSA (~2.4 Å)

are slightly smaller than that in the absence of BSA ($\sim 2.6 \text{ \AA}$) (Supplementary Fig. 4(a)). ϵ in the inhibitor-unbound state are also not affected by the presence of the BSAs (Supplementary Fig. 4(b)). Something is missing in this sentence.

11- Figure S8

The authors could compare the binding sites of PP1 on BSA with the binding sites of other drugs (see, for instance, PDB 4JK4).

12- Page 8, line 143: 'independent assays' – independent.

13- Consider swapping Figure 2A with Figure S10, which makes clear that, in the presence of many proteins, the probability of PP1 being bound to a protein is proportional to the SASA of the protein.

14- Table S6

Explain the meaning of the terms in the equation.

15- Page 9: 'This binding pathway is different from that observed in Src8BSA: PP1 first reaches the hinge region of the kinase and then intrudes into the canonical binding site.'

This is true for 1/2 cases in figure S13. Please rewrite it.

16- Pages 9-10, lines 176-190

The argument to be made is that the simulation without restraints is equivalent to the simulation without crowder, while the simulation with restraints on kinase is equivalent to the simulation in the presence of crowder, which limits the motions of kinase, but this is not really apparent in the paragraph. Please rewrite it.

17- Page 10, lines 191-205

Repetition from the previous paragraph.

18- Page 11, line 206: 'Conformatioal shifts' – conformational.

19- Page 11-12, section 'Conformatioal shifts of a Tyr sidechain upon crowding'.

The conclusions about Tyr82 are interesting, but I would expect that both paths (hinge region and G-loop) would be observed in crowded conditions. This is probably a sampling problem, as only 2 binding events were observed in crowded conditions. The authors should comment about it in the discussion.

20- What is the conformation of Tyr82 in the apo form of kinase? It seems it is Tyr-out in figure S18 (in which the residues should be labelled).

Does the Tyr-in conformation appear in any crystal structure? Could the Tyr-in conformation be a force field artifact?

21- Page 3 of supporting information, section 'Definition of reaction coordinates for free energy landscapes'.

The definition of the five anchor atoms (L0, L1, P1, P2, and P3) is very confusing. Does it change according to trajectory or frame? It would be easier to give the identity of each anchor atom with the definition.

Reviewer #3 (Remarks to the Author):

In their manuscript (NCOMMS-20-35226) Sugita, Feig, and co-workers use extensive modeling and analyses, and experimental measurements to demonstrate that crowding reduces efficacy of a kinase

inhibitor. The study is based on extensive and well-designed set of all-atom simulations of systems containing c-Src kinase, small inhibitor (PP1), and bovine serum albumin (BSA), which is present at varying concentrations and serves as a crowder. The study is detailed and findings are well supported.

The all-atom molecular dynamics trajectories captured spontaneous PP1 binding into the active site of c-Src kinase and the authors were able to show that crowding changes the binding pathways as in the presence of BSA PP1 concentration is reduced near the kinase. It is insightful that the authors were able to provide mechanistic explanation of the importance of excluding volume and of weak non-specific PP1-protein interactions. Importantly, the computational findings are supported by experimental binding data.

These findings have implications for future studies of drug interactions in native-like environments and will be of great interest to the readers of Nature Communications. The manuscript should be published mostly as is; I have found only a few places in the text which require edits related only to formatting:
Line 106 "e in the inhibitor-unbound" has missing text.
Line 175: The paragraph is repeated starting from the Line 190.

Reviewer #1 (Remarks to the Author):

The manuscript is an important contribution and an exciting piece of science. Since the pioneering computational efforts carried out by the manuscript is an important contribution and an exciting piece of science. Since the pioneering computational efforts carried out by the authors teams (M. Feig's and Y. Sugita's), this new work is a beautiful example showing that the modelling of the macromolecular crowding is an essential aspect to consider in order to understand in vivo biological processes.

The manuscript is clear, well written and presented. The technical aspects are neat. The in-silico results, obtained with great statistics considering the systems, are complemented by ad hoc experiments that try to reinforce the main conclusion of the investigation. I consider the manuscript publishable in Nature Communications, but several aspects need to be addressed prior to publication. The points are listed below.

(1) My first aspect concerns the “sequestration” thermodynamics of the inhibitor that is at the origin of the trapping mechanism of the molecule on the surface of the crowders. This interaction limits the independent diffusion of the molecule in the solvent ‘phase’, producing a strong slowdown that in turn quenches the access to the target protein. It will be important to report/show/stress that within the used force field the solvation free energy of the ligand and the interaction free energy with representative surface patches of the crowder BSAs are not unbalanced artificially. The authors can easily produce these data by independent calculations.

Molecular mechanics-generalized Born surface area (MM-GBSA) analysis on the binding affinity of PP1 to either c-Src kinase or BSA was performed to address the concern. We confirmed the interactions of PP1 with c-Src kinase and with BSA are overall “well balanced”, i.e. the calculated binding affinities to different surface regions are of similar magnitude between the two proteins and between different surface patches. The canonical binding site for PP1 in c-Src kinase is found at the lowest binding free energy. There are several similarly or even slightly more favorable binding sites on BSA as well, but it is in fact the point of this study that off-target ligand binding may significantly distract from on-target binding events under crowded conditions. The binding free energies at different sites are shown in Supplementary Fig. 9. In addition, following sentences were added at the end of the page 7 in the revised manuscript:

“This observation is consistent with the PP1 binding free energies at different binding sites on c-Src kinase and BSA estimated through molecular mechanics-generalized Born surface area (MM-GBSA) analysis (Supplementary Fig. 9). The range of binding free energies at different c-Src kinase-PP1 binding site are comparable to those at BSA-PP1 binding sites. Due to the crowding,

the probability of finding PP1 in the water phase is significantly reduced (Supplementary Fig. 10, Supplementary Table 4)”

(2) *Strictly related to the point mentioned above, the authors should better discuss and account for the discrepancy between the simulated and the experimental conditions. In the experiments, they only add 5 g/L of BSA (which corresponds to a very small volume fraction of ~ 0.004 , right?). So, to observe any strong difference in the bulk concentration of the inhibitor, which they suggest as a main rationale of the decreased inhibitor efficiency, the inhibitor should bind very strongly to BSA. However, in the simulations, they only observe a drop in the bulk concentration of the inhibitor from 80 % in dilute conditions to 30 % at a volume fraction of 0.11. This says that for the much smaller experimental volume fraction, the bulk concentration should be very similar to the dilute case. Therefore, the sequestration would not be sufficient for explaining the experimentally detected order-of-magnitude shift in inhibitor concentration.*

We additionally carried out a new experiment increasing BSA concentration up to 100 mg/ml (Fig. 3). This concentration corresponds to the concentration in the Src2BSA simulation. With the high BSA concentration, the inhibition of PP1 becomes much milder. On page 9 in the revised manuscript, we added the sentences:

“We also performed the same inhibition assays at the higher concentration of BSA (100 mg/ml). In the experiments, the IC_{50} of PP1 further increased to 12.2 μ M, indicating even weaker inhibition by PP1 in the presence of higher concentrations of BSA.”

The experimental conditions of the old and new experiments are the same except for the concentration of DMSO in the buffer solutions (0.2% and 1% in the old and new experiments). We explained these differences between the first and the second experiments on page 17:

“The second *in vitro* inhibition assay was performed in the same conditions except for the concentrations of BSA (100 mg/ml) and DMSO (1.0%).”

(3) *On the basis of the sequestration on crowders surfaces and the associated slowdown dynamics of the inhibitor, the authors open a discussion about drug design strategy in order to account for the realistic environmental conditions of the cell interior. Here I have a question. Looking at the experiments reported in Fig. 2 bottom panel it seems that in order to recover the inhibition efficiency observed in dilute condition it suffices to increase the inhibitor concentration (a factor of 10). In drug design and subsequent essays, the concentration is always a free*

parameter that needs to be tuned to grant efficiency and avoid toxicity. If the author enters in this arena, they should for instance discuss what concentration limits should be avoided. Is there an example that can be reported? Given the great experience cumulated with past simulations (see the authors past works), is there a rule of thumbs that relates slowdown motion & local sequestration and crowder concentration so to estimate the possible shift in drugs concentration needed to recover dilute solution efficiency?

We believe that our work further supports the qualitative assertion that a consequence of cellular crowding is an effectively lower inhibitor concentration due to reduced activity and diffusion, but it may require further efforts to arrive at the more quantitative ‘rule of thumb’ that the reviewer is asking for. One issue is the large time-scale gap between simulations and experiments that limits exact quantitative predictions. To address this challenge, we have to accelerate biological events in simulations at higher crowder concentrations in order to be able to study ligand binding events. We also generally consider higher macromolecule concentrations in the simulations than experiments to reduce computational costs, which again is limiting the ability to make exact quantitative predictions until now.

(4) Finally, a personal view. Considering that great effort in drug design is spent to refine the chemistry of the active molecule to grant high affinity with target binding sites, and that basically the slowdown/sequestration caused by the environment seems not highly specific, I think very difficult to incorporate these two requirements (highly affinity for a target, poor interactions with general protein surfaces) at once. Have the authors some ideas? A finest analysis of the inhibitor with the crowders, that is not present in the manuscript and should, could provide some suggestions. I think also the authors should add in the introduction a note on the effect of diffusion in molecular recognition under crowding. The competitive effect of local high concentration and slowdown motion reported in the “Discussion” section, can be already anticipated in the introduction.

First, we added sentences about the crowding effect of diffusion in Introduction, on page 3 of revised manuscript, according to the suggestion of the reviewer:

“Diffusion of a ligand toward a target protein is generally reduced, while the transition-state stabilization and/or encounter-complex formation are facilitated, as the protein concentration increases. These two effects make it difficult to understand the overall effect of crowding on enzymes⁴.

As the reviewer pointed out, it is not straightforward to satisfy the two requirements (high affinity

for a target and poor interactions with general protein surfaces) in the current drug discovery strategy. More quantitative ligand-binding calculations, for instance, free-energy calculations, may help to look for such a drug candidate. Since we have already mentioned such a future work on page 15 in the original manuscript, we did not change the revised manuscript for this comment.

“In future work, binding free energy calculations under crowded conditions could quantitatively address the relation between the effective inhibitor concentration and its efficacy.”

The reviewer is also asking for a more extensive analysis of the inhibitor with crowders. This is an excellent point but in a qualitative sense we expect to find well-known patterns of interaction (shape complementarity, electrostatic hot spots) which, we believe may not add much here. We did however, also in response to the second reviewer, compare binding of PP1 to BSA with reported binding poses of other compounds. The finding is that they overlap to some degree.

It would be of greater value to carry out a thorough analysis of how exactly inhibitor binding to the crowders differs from binding to the kinase and binding to BSA. We think that this is a good question for future work and in some sense separate from the crowding simulations presented here because it can be addressed more efficiently by comparing inhibitor binding to a single molecule of Src-kinase vs. a single molecule of BSA. Such a study should also consider a wider range of other ‘crowder’ proteins in order to be able to come to more general conclusions. We believe that such work is clearly out of the scope of the present study.

(5) One extra remark concerns the discussion of the conformational changes induced by the BSAs crowders and the altered distribution of the Y82-G86 distance, a proxy for the conformational state of Tyr82. To me, the text seems light on this. The authors observe a change but I do not find a connection with the presence of the crowders. I think this must be better described/analyzed and a structural effect indicated, or at least queried.

We have tried to figure out the connection between the Tyr82 conformations and the crowding conditions. The C α atom contacts between c-Src kinase and BSA in each simulation were examined for TYR-in and TYR-out conformations, individually (Supplementary Figure 24). However, no significant differences appeared between the two conformations. This suggests the effect of crowders on the local protein conformation could be more complicated, and we have to leave this question for future work. To mention this point, we added the following sentence to the Discussion on Page 15.

“To examine the difference of c-Src kinase-BSA interactions between TYR-in and TYR-out conformations, we calculated the average number of the C α atom contacts between BSA and c-Src kinase for the two conformations individually (Supplementary Figure 24). However, no significant differences between the two conformations is observed, suggesting that the crowder effects on the local protein conformation could be more complicated.”

(6) *One delicate aspect the manuscript does not address is the homogenous vs heterogenous crowding effect. Since we know that big vs small crowders can have different effects on local packing, stability, and conformational fluctuations, and since the change in probability of Tyr82 is visible even in presence of two BSAs, the authors could, with a small effort, produce a simulation with smaller crowders, or replacing 1 BSA with a different crowder. This would reinforce the manuscript and the results.*

We have recognized that various properties of solute proteins are changed depending on the crowder sizes from the past experimental studies, for instance, ref. 6 (Miklos et al. JACS 133, 7116-7120 (2011)). However, to simulate different crowding systems is in fact not a small effort for us even if just one BSA in the present systems is replaced by a different crowder. We note that the simulations presented here are based on rather extensive computational resources including simulations on the special-purpose Anton2 supercomputer where our access to resources is very limited. While we would like to address this question, it will need to remain a topic of future work. In the revised manuscript on page 16, we just added the following sentences added in Discussion.

“The effects of different crowder types, for instance, the size of crowders⁶ on protein structures and stability were investigated experimentally. The intracellular environments indeed consist of heterogeneous proteins and other biomolecules, as we simulated in the previous simulation study¹⁵. A further investigation using heterogenous crowders along the present study would lead to a deeper understanding of how the local conformational changes in more realistic cellular environments alter the binding mechanism.”

(7) *Final curiosity. It is intriguing that binding events are visible only in the simulations of the dilute system and at the highest crowding concentration, and not at intermediate values of crowding. See Table 1 in SI. Why? This must be discussed!*

This is an interesting suggestion about further discussion in the paper. As shown in Fig. 5, the dilute system emphasizes the Tyr-in conformation, while the highest crowding concentration increases the population of Tyr-out. One hypothesis is that these two extreme cases increase the

ligand binding possibilities through different pathways. If we can perform our MD simulations much longer, more binding events through either one of the pathways would be observed in Src2BSA and Src4BSA systems, since the conformational changes seem to happen due to the population shifts. However, it is currently difficult using our available computer resources. In this study, we already carried out extensive conformational samplings using very significant computer time on supercomputers as well as the MD-specific supercomputer, Anton2. We added the discussion pointed out by the reviewer on page 15.

“In the study, we observed binding events only in the dilute and Src8BSA systems, which are considered as two extreme conditions. These two conditions may increase the possibility of binding processes on different pathways. Since the conformational changes between Tyr-in and Tyr-out happen as population shifts (Fig. 5), longer MD simulations in Src2BSA and Src4BSA could increase the possibilities of ligand-binding processes through one of the two pathways.”

(8) *The paragraph reporting on binding pathway is present twice in the manuscript (pg 9 lines 176-190, and pg 10 191-205). A cut and paste error, I guess.*

We are sorry for our mistake. We removed the paragraph in the revised manuscript.

(9) *Fig. 2. The volume fraction associated to the experimental condition, BSA 5 mg/ml, should be indicated so to have a link between the top and the bottom panels.*

Since the experimental condition (BSA 5 mg/ml) is two order different from the simulation conditions, it is difficult to link the top and bottom panels in the original Fig. 2. This issue was pointed out also by reviewer #2. By following the suggestion by reviewer #2, we first split the original Fig. 2 into Figs. 2 and 3 for discussing the results separately in the revised manuscript and then swapped the original Fig. 2 (top) with Supplementary Figure 10, which is now shown as Fig. 2 in the revised manuscript. The original Fig. 2(top) is shown as Supplementary Fig. 10 in the revised SI file. To link more between experiments and simulations, we added a new experiment at the higher BSA condition (100 mg/ml) and showed the results in Fig. 3.

(10) *Fig. 3 can be improved by labeling explicitly panels c and d, with the “dilute” and “Src8BSA” labels so to ease a reader to spot the differences immediately.*

The figure 4 in the revised manuscript was modified accordingly.

Reviewer #2 (Remarks to the Author):

The authors investigated the diffusion of the low molecular weight inhibitor PP1 and its binding to Src kinase in the absence and presence of crowding. Crowding is mimicked by the use of different concentrations of the crowder bovine serum albumin. The authors suggest, based on experiments, that PP1 is less efficient as an inhibitor in the presence of crowders. They also observe that PP1 is less available in solution with increasing crowder concentration, and that crowders may change the pathway for ligand binding due to conformational changes in Tyr82 of Src kinase.

The manuscript is generally clearly presented and mostly well written. The manuscript addresses a significant problem for which molecular simulations can give important insights. The results are original and interesting. However, I have some major concerns as detailed below, some details are missing in the methods, and the conclusions are not fully supported by the results.

Major concerns

(1) Low sampling of binding pathway

Page 14, line 269: 'In Src8BSA simulations, the PP1 binding pathway is entirely different from that in dilute simulations.'

The authors obtained very few binding events in the simulations, especially in crowded conditions (4 and 2 events in the absence and presence of crowding, respectively). Therefore, they would need to sample more binding events to be able to claim that binding pathways are indeed different in the absence and presence of crowding.

We performed the additional 30 simulations (each for 20 ns) starting from either **E** state (G-loop) or **E_r** state observed in the previous Src8BSA simulations. Three binding events from **E_r** state were newly observed, while no events from **E** states (Supplementary Figure 17). Thus, we can claim that the crowder proteins alter the binding pathways. Following sentences were added in the subsection "Ligand-binding pathways in crowded environments" (on Page 11).

"To examine if the difference happened to be observed, we performed additional 30 simulations (each for 20 ns) in Src8BSA starting from **E** and **E_r** states. While the simulations from **E_r** state lead to three binding events, no binding was observed in the simulation from **E** state. Similar to the binding trajectories shown in Fig. 4(d) and Supplementary Fig. 16, the binding trajectories from **E_r** state fit to 'restraint' FEL (Supplementary Fig. 17). Thus, it can be safely stated that the binding pathways are altered in the presence of crowders."

(2) *Experimental results*

The plot in Fig 2a shows computed values that do not directly relate to the main heading of the figure as a function of crowder protein volume fraction. The corresponding experimental value for the protein volume fraction would be at less than 1% and at a different ratio of kinase to crowder. Thus, this plot has little relation to the experimental data in Fig 2b and I think it would be better to put these two plots into different figures.

Based on the reviewers' suggestion, we split the original Fig. 2 into the new Fig. 2 (for simulation) and Fig. 3 (for experimental data) in the revised manuscript. Also, as responded to the reviewer 1, we carried out new experiments to collect data at increased BSA concentrations (Fig. 3).

Page 8, lines 151-153: 'The 50% inhibition concentration (IC₅₀) of the PPI considerably increased from 33.80 nM to 290.4 nM in the presence of BSA (Fig. 2(b)). This validates our hypothesis of a reduced inhibitor efficacy in crowded protein solutions.'

Here, the authors need to point out that the enzyme activity is higher in the presence of BSA and give what the reference was for defining 100% enzyme activity. If the reduced inhibitor efficacy is due to differences in inhibitor-protein binding, one wonders why the same effect is not observed for substrate; indeed, the data suggest the opposite. However, at no point is it stated what the Src substrate measured actually is (1309, p15). This information needs to be added, along with data on the dependence of Src substrate binding on BSA crowding, e.g. from analogous simulations to those for the inhibitor.

In the current simulation study, we did not include Src substrate, while, in the experiments shown in Fig. 3, we examined the enzymatic activity of c-Src kinase with ATP and Src substrate in the presence or absence of BSA as crowders. Although the dependence of Src substrate binding on BSA crowding is interesting theoretically and experimentally, analogous simulations to those for the inhibitors are greater effort for us than the current ones. We thank the reviewer to suggest this interesting future research and would like to try it in our future works.

(3) *Incomplete description of computational methods*

The setup and parameters for simulations are well described, but there is no information (in the manuscript or in the supporting information) about how the analysis of these simulations was performed (calculation of spatial distribution functions, free energy landscapes, and mean square displacements; definition of the bound state). This should be added.

We added the details of computational methods in the supplementary information as “Analysis Details” (Page 3 and 4).

(4) *Page 2, lines 44-45: 'Protein functions in a living cell could be examined using atomistic MD simulations with realistic cellular environments.'*

This is an overstatement, compared to what is presented in the paper and must be removed or edited. The interaction with one artificial crowder, BSA, is studied.

We follow the reviewers' opinion. The scope of our paper is indeed limited by what we can achieve with current computer methodology. In the revised manuscript, we removed the sentence.

Minor concerns

(5) *Abstract: Give the full name of PP1. It only becomes clear what it is in Supplementary Figure 2 and this figure requires a pdb identifier and a citation for the crystal structure. 'imposed between the highlighted carbon atoms in PP1s'. Only 1 carbon is highlighted by a red circle. Rephrase this sentence to avoid confusion.*

We combined Supplementary Figs. 1, 2 and 5 as Supplementary Fig. 1. The caption of the figure was modified according to the comment as follows.

“(a) The crystal structure of the c-Src kinase-PP1 inhibitor complex. The structure was built using the X-ray structures of the c-Src kinase (PDB ID: 1Y57) and of the autoinhibited form of Hck complexed with PP1 (PDB ID: 1QCF). (b) A close up view of the canonical binding site (ATP binding site) with bound PP1. The circled residues (Glu81, Thr80, and Met83) form the hydrogen bonds to PP1 (c) The residues 100-150, 168-200, and 225-259 (colored in purple) in MD simulation snapshots used to superimpose to those in the crystal structure in the analysis of root mean square fluctuation (RMSF) and RMS displacement (RMSD). (d) Definition of the protein-PP1 distance, ξ .”

(6) *Page 4, line 77: 'Atomic MD simulations' ; change to atomistic or atomic detail >*

Thank you. The word ‘Atomic’ was changed to ‘atomistic’.

(7) *Page 4, lines 89-90: 'Dysregulation of this kinase function is associated with many diseases like cancer, making it an important therapeutic target.'*

A reference for this statement is missing.

The following references were cited as 40 and 41.

40. Zhang, J., Yang, P. L. & Gray, N. S. Targeting cancer with small molecule kinase inhibitors. *Nat. Rev. cancer* **9**, 28 (2009).

41. Ferguson, F. M. & Gray, N. S. Kinase inhibitors: the road ahead. *Nat. Rev. Drug Discov.* **17**, 353 (2018).

(8) *Page 4, lines 95-96: 'We performed atomistic MD simulations of four c-Src kinase/PPI/BSA systems, each of which contains 0 (dilute solution), 2, 4, and 8 BSAs in a simulation box'.*

What would be the corresponding concentration of BSA in the simulations, in g/L? This should be added to Supplementary Figure 3.

We added the information on BSA concentrations in g/L for each system in Supplementary Table 3 (Page 3).

(9) *Page 5, line 105: 'The C α root mean square deviations (RMSDs) from the X-ray structure are observed around 2-3 Å'; in all the simulations, indicating that the kinase remains in the active conformation'.*

A small RMSD does not indicate whether the kinase is in the active conformation or not. The authors should examine the residues in the DFG motif to make this statement.

We analyzed the pseudo-torsion angles related with DFG motif proposed by Möbitz (Biochim. Biophys. Acta, **1854**, 1555 (2015), cited as Ref. 53 in the main text). The result is shown in Supplementary Figure 4. We confirmed that the kinase in our simulations kept the active-like initial conformation (DFG-in and α C-out state). Following sentence was added in 'Structural stability and flexibility' (Results) in the main text (Page 5, line 108).

“Two pseudo-torsion angles of Asp-Phe-Gly (DFG) motif (Supplementary Fig. 4), which represent active/inactive states of the kinase⁵³, show that the kinase remains in the active-like initial conformation (DFG-in and α C-out state) during the simulations.”

In addition, we added the definition of the pseudo-angles in section “Analysis Details” in supporting information as “The classification of DFG-in/DFG-out states of c-Src kinase”. (Page 4, line 7).”

(10) Page 5, lines 106-108: *The time averages of the RMSD values in the presence of BSA (~2.4 Å) are slightly smaller than that in the absence of BSA (~2.6 Å) (Supplementary Fig. 4(a)). e in the inhibitor-unbound state are also not affected by the presence of the BSAs (Supplementary Fig. 4(b))*’.

Something is missing in this sentence.

The sentence was modified as follows (Page 5, line 106).

“The C α root mean square fluctuations (RMSFs) of the kinase in the inhibitor-unbound state are also not affected by the presence of the BSAs ... (abridged) ...”

(11) Figure S8

The authors could compare the binding sites of PP1 on BSA with the binding sites of other drugs (see, for instance, PDB 4JK4).

We add the figure of the crystal structure 4JK4 and the following sentence was added (Page 7, line 143).

“The binding sites of PP1 on BSA resemble those of other compounds found in the crystal structure (PDB ID: 4JK4), reflecting the general feature of BSA as a vehicle of small molecules in the circulatory system (Supplementary Fig. 7(b))⁵⁶”

(12) Page 8, line 132: *‘independent assays’; independent.*

The mistake was corrected.

(13) Consider swapping Figure 2A with Figure S10, which makes clear that, in the presence of many proteins, the probability of PP1 being bound to a protein is proportional to the SASA of the protein.

The figures were swapped.

(14) Table S6

Explain the meaning of the terms in the equation.

The explanation was added in “Analysis Details” in supporting information as “Residence time

correlation functions of PP1” (Page 5).

(15) Page 9: *'This binding pathway is different from that observed in Src8BSA: PP1 first reaches the hinge region of the kinase and then intrudes into the canonical binding site.'*

This is true for 1/2 cases in figure S13. Please rewrite it.

The sentence was modified as follows (Page 10, line 195).

“PP1 first makes a contact with either the N-terminal side of G-loop or hinge region (Supplementary Fig. 13) and then intrudes into the canonical binding sites.”

(16) Pages 9-10, lines 176-190

The argument to be made is that the simulation without restraints is equivalent to the simulation without crowder, while the simulation with restraints on kinase is equivalent to the simulation in the presence of crowder, which limits the motions of kinase, but this is not really apparent in the paragraph. Please rewrite it.

The following sentences were added on Page 11 of the revised manuscript.

“The binding trajectories in dilute solution fit to the major pathway of ‘free’ FEL (Fig. 4(c)) and those in Src8BSA fit to the major pathway of ‘restraint’ FEL (Fig. 4(d)). The major difference in two pathways is the interaction at an initial encounter state: PP1 interacts with the G-loop in dilute solution (E) (Supplementary Fig. 12), while it interacts with the N-terminal side of the G-loop or hinge region in Src8BSA (Er) (Supplementary Fig. 13). To examine if the difference happened to be observed, we performed additional 30 simulations (20 ns each) in Src8BSA starting from E and Er states. While the simulations from Er state lead to three binding events, no binding was observed in the simulation from E state. Similar to the binding trajectories shown in Fig. 4(d) and Supplementary Fig. 16, the binding trajectories from Er state fit to ‘restraint’ FEL (Supplementary Fig. 17). Thus, it can be safely stated that the binding pathways are altered in the presence of crowders.”

(17) Page 10, lines 191-205

Repetition from the previous paragraph.

The paragraph was removed.

(18) Page 11, line 206: 'Conformatioal shifts'; conformational.

The mistake was corrected.

(19) Page 11-12, section 'Conformatioal shifts of a Tyr sidechain upon crowding'.

The conclusions about Tyr82 are interesting, but I would expect that both paths (hinge region and G-loop) would be observed in crowded conditions. This is probably a sampling problem, as only 2 binding events were observed in crowded conditions. The authors should comment about it in the discussion.

We performed the additional 30 simulations (each for 20 ns) starting from either **E** state (G-loop) or **E_r** state in Src8BSA. We obtained three binding events from **E_r** state, while no events from **E** states (Supplementary Figure 17). Thus, we can claim that the crowder proteins alter the binding pathways. Following sentences were added in the subsection “Ligand-binding pathways in crowded environments” (Results on Page 11).

“To examine if the difference happened to be observed, we performed additional 30 simulations (20 ns each) in Src8BSA starting from **E** and **E_r** states. While the simulations from **E_r** state lead to three binding events, no binding was observed in the simulation from **E** state. Similar to the binding trajectories shown in Fig. 4(d) and Supplementary Fig. 16, the binding trajectories from **E_r** state fit to ‘restraint’ FEL (Supplementary Fig. 17). Thus, it can be safely stated that the binding pathways are altered in the presence of crowders.”

In addition, as the reviewer pointed out, since the both TYR-in/out conformations are present in all the systems (Fig. 5), the binding events could occur from both the pathways in all the systems if we extend the simulation length. Hence, we added the following sentences in Discussion (Page 14, line 285).

“Since the conformational changes between Tyr-in and Tyr-out happen as population shifts (Fig. 5), longer MD simulations in Src2BSA and Src4BSA could increase the possibilities of ligand-binding processes through one of the two pathways.”

(20) *What is the conformation of Tyr82 in the apo form of kinase? It seems it is Tyr-out in figure S18 (in which the residues should be labelled).*

Does the Tyr-in conformation appear in any crystal structure? Could the Tyr-in conformation be a force field artifact?

In the revision, we additionally performed 1 μ s simulations for dilute and Src8BSA with different force field, CHARMM36m. Similar to the original results using AMBER forcefield, the population shift from TYR-in to TYR-out conformations is observed upon the crowding. As for the apo form crystal structures, TYR-out conformation appears in most cases, but several kinases have TYR-in conformation (for instance, protein kinase B/Akt). Following sentences were added. ('Conformational shifts of a Tyr sidechain upon crowding' (Results), on Page 13)

"The additional 1 μ s-simulations using a different force field, the CHARMM36m forcefield⁵⁸, support that the observed population shift between TYR-in and TYR-out is universal (Supplementary Fig. 18)."

(Discussion on Page 16 in the revised manuscript)

"Most of the apo form crystal structures of kinases, including c-Src kinase (PDB ID: 1YOJ)⁶⁸, have TYR-out conformation, although there are crystal structures having TYR-in conformation (such as protein kinase B/Akt⁶⁹)."

(21) *Page 3 of supporting information, section 'Definition of reaction coordinates for free energy landscapes'.*

The definition of the five anchor atoms (L0, L1, P1, P2, and P3) is very confusing. Does it change according to trajectory or frame? It would be easier to give the identity of each anchor atom with the definition.

The atoms used for defining anchor atoms are determined from the crystal structure, and hence the anchor atoms do not change according to trajectory. We gave the identity of each anchor atom in the supporting information ('Definition of reaction coordinates for free energy landscapes' in Analysis Details, Page 3)

Reviewer #3 (Remarks to the Author):

In their manuscript (NCOMMS-20-35226) Sugita, Feig, and co-workers use extensive modeling and analyses, and experimental measurements to demonstrate that crowding reduces efficacy of a kinase inhibitor. The study is based on extensive and well-designed set of all-atom simulations of systems containing c-Src kinase, small inhibitor (PP1), and bovine serum albumin (BSA), which is present at varying concentrations and serves as a crowder. The study is detailed and findings are well supported.

The all-atom molecular dynamics trajectories captured spontaneous PPI binding into the active site of c-Src kinase and the authors were able to show that crowding changes the binding pathways as in the presence of BSA PPI concentration is reduced near the kinase. It is insightful that the authors were able to provide mechanistic explanation of the importance of excluding volume and of weak non-specific PPI-protein interactions. Importantly, the computational findings are supported by experimental binding data.

These findings have implications for future studies of drug interactions in native-like environments and will be of great interest to the readers of Nature Communications. The manuscript should be published mostly as is; I have found only a few places in the text which require edits related only to formatting:

(1) Line 106 “in the inhibitor-unbound” has missing text.

The sentence was modified as follows (Page 5, line 106).

“The Ca root mean square fluctuations (RMSFs) of the kinase in the inhibitor-unbound state are also not affected by the presence of the BSAs ... (abridged) ...”

(2) Line 175: The paragraph is repeated starting from the Line 190.

The paragraph was removed.

REVIEWER COMMENTS

Reviewer #1 (Remarks to the Author):

The manuscript is ready for publication. The authors addressed all the points raised, and the manuscript is improved.

Reviewer #2 (Remarks to the Author):

The authors have greatly improved the manuscript. In particular, the description of the setup of simulations and the analysis of the trajectories is much clearer. They have also added additional simulations. However, I still have the following concerns.

Major concerns

1- Low sampling of binding pathways

The authors have added new simulations, and these support the differences in the presence of crowders. However, the sampling remains low: 3 binding events vs 0 binding events to distinguish the cases. Because of this, the conclusions drawn needs to be worded more cautiously in the manuscript.

2- Experimental results

It is good that an additional point at a higher BSA concentration was added to the inhibition measurements. However, the conditions were changed to increase the concentration of DMSO. This increase is understandable but requires an additional control experiment at the higher DMSO concentration (5 times higher than the original concentration) in the absence of BSA so that the effects of BSA and DMSO can be distinguished. This experiment is currently lacking.

Furthermore, the fitting lines in Fig 3 have been added compared to the previous version of the manuscript. The fits for the measurements with BSA do not look valid given the data points shown. The initial 100% point is not given by the 5mg/ml data. It looks like the authors have changed the meaning of the y-axis between versions but this is not described in the legend or text. The 100mg/ml plot has a minimum enzyme activity of 60% which is obviously high for computing an IC50.

The authors have not added information to the manuscript on the identity of the Src substrate. While I understand the opinion of the authors that simulations of the binding of the substrate would go beyond the scope of the present simulation study, I do not think that the substrate can be completely ignored. The simulations pertain only to binding. It is not possible to interpret the IC50 data with respect to inhibitor binding without further information on the assay done.

Reviewer #3 (Remarks to the Author):

In their revision of the manuscript (NCOMMS-20-35226) Sugita, Feig, and co-workers have satisfactorily addressed all my comments and suggestions. The manuscript represents a novel, significant, and insightful contribution that advances the field. It will be of great interest to the readers of Nature Communications, and can be published as is.

Reviewer #2 (Remarks to the Author):

The authors have greatly improved the manuscript. In particular, the description of the setup of simulations and the analysis of the trajectories is much clearer. They have also added additional simulations. However, I still have the following concerns.

Major concerns

1. Low sampling of binding pathways

The authors have added new simulations, and these support the differences in the presence of crowders. However, the sampling remains low: 3 binding events vs 0 binding events to distinguish the cases. Because of this, the conclusions drawn needs to be worded more cautiously in the manuscript.

We modified the sentence in the end of section “Ligand-binding pathways in crowded environments” on page 11 as follows:

“This suggests that the binding pathways might be altered in the presence of crowders.”

In addition, the following sentences were added in Discussion on page 15:

“Since we only perform the conventional MD simulations, the sampling of the binding events is still low. Further investigation with an enhanced sampling technique in the presence of the crowders could further clarify the change of the pathway due to the crowding.”

2. Experimental results

It is good that an additional point at a higher BSA concentration was added to the inhibition measurements. However, the conditions were changed to increase the concentration of DMSO. This increase is understandable but requires an additional control experiment at the higher DMSO concentration (5 times higher than the original concentration) in the absence of BSA so that the effects of BSA and DMSO can be distinguished. This experiment is currently lacking.

To reply this comment, we carried out the enzyme assays again, adding a control experiment at the higher DMSO concentration. We show the enzyme activities in dilute and BSA (5 mg/ml) solutions at the low DMSO concentration (0.2%) in Figure 3a and those in dilute and BSA (100 mg/ml) solutions at the higher DMSO concentration (1.0%). As expected by this Reviewer, the

changes of DMSO concentration affects slightly the enzyme activities of c-Src kinase. However, the effect of BSA crowder concentrations is much greater. We are thus able to distinguish the effects of BSA and DMSO on the enzyme activity in the revised manuscript.

Furthermore, the fitting lines in Fig 3 have been added compared to the previous version of the manuscript. The fits for the measurements with BSA do not look valid given the data points shown. The initial 100% point is not given by the 5mg/ml data. It looks like the authors have changed the meaning of the y-axis between versions but this is not described in the legend or text.

Both the previous and current version of the manuscript define the 100% enzyme activity as the activity in the absence of PP1 inhibitor. The difference between the versions stems from whether the constraint is imposed on the fitting so as to satisfy the enzyme activity of 100% in the absence of PP1 or not. In the current version, the constraint is imposed. On pages 17-18, we added the following sentence:

“The enzyme activity in the absence of PP1 was set to 100% in each condition. IC₅₀ was determined by curve fitting the data using the GraphPad Prism8 program (GraphPad software). The fitting function is given by

$$(\text{Enzyme activity}) = \frac{100}{1 + (\text{IC}_{50}/[\text{PP1}])^\alpha},$$

where α is Hill coefficient. IC₅₀ and α are used as the fitting parameters, and the constraint was imposed on the fitting so as to satisfy the enzyme activity of 100% in the absence of PP1.”

The 100mg/ml plot has a minimum enzyme activity of 60% which is obviously high for computing an IC50.

As pointed out by the reviewer, we only performed the experiments in which the concentration of PP1 inhibitor is lower than the computed IC₅₀ in 100 mg/ml condition. This is because PP1 is not soluble at [PP1] > 10 μ M. After reperforming the experiment, we got more accurate results with less standard deviation (Figure 3b). Then, the IC₅₀ is estimated from the interpolation of the fitting curve. We believe that the concentration effect of the crowders on the enzyme activity

can be already seen from the current our experiments shown in Fig. 3. To address the reviewer's comment, we added the following sentence on page 9:

“In this condition, the minimum enzyme activity was reduced to ~30% at [PP1] = 10 μ M and higher concentrations of PP1 were difficult to achieve because of limited solubility of PP1.”

The authors have not added information to the manuscript on the identity of the Src substrate. While I understand the opinion of the authors that simulations of the binding of the substrate would go beyond the scope of the present simulation study, I do not think that the substrate can be completely ignored. The simulations pertain only to binding. It is not possible to interpret the IC50 data with respect to inhibitor binding without further information on the assay done.

Thank you for pointing out the missing information in the manuscript. The Src substrate used in the assay is KVEKIGEGTYGVVYK-amide (SignalChem Pharmaceuticals). This information was added in Methods section on page 17.

REVIEWERS' COMMENTS

Reviewer #2 (Remarks to the Author):

The authors have adequately addressed my concerns.